# Waveform Design for Improved Detection of Extended Targets in Sea Clutter

**DOI:** 10.3390/s19183957

**Published:** 2019-09-13

**Authors:** Linke Zhang, Na Wei, Xuhao Du

**Affiliations:** 1Key Laboratory of High Performance Ship Technology, Wuhan University of Technology, Ministry of Education, Wuhan 430063, China; lincol@whut.edu.cn; 2Key Laboratory of Marine Power Engineering & Technology, Wuhan University of Technology, Wuhan 430063, China; 3Key Laboratory of Modern Acoustics and Institute of Acoustics, Nanjing University, Nanjing 210000, China; 4Department of Mechanical Engineering, The University of Western Australia, Crawley 6008, Australia; xuhao.du@uwa.edu.au

**Keywords:** waveform design, cognitive radar, sea clutter, extended target, low range sidelobes

## Abstract

Adaptive waveform design for cognitive radar in the case of extended target detection under compound-Gaussian (CG) sea clutter is addressed. Based on the CG characteristics of sea clutter, the texture component is employed to characterize the clutter ensemble during each closed-loop feedback and its estimation can be used for the next transmitted waveform design. The resulting waveform design problem is formulated according to the following optimization criterion: maximization of the output signal-to-interference-plus-noise ratio (SINR) for sea clutter suppression, and imposing a further constraint on sidelobes level of the waveform autocorrelation outputs for decreasing the false alarm rate. Numerical results demonstrate the effectiveness of this approach.

## 1. Introduction

Radar detection in an ocean environment is particularly challenging due to the non-Gaussian traits of sea clutter returns as viewed by high-resolution radars and/or at low grazing angles [1]. Most traditional studies only focus on adaptive detection algorithms at the receiver end, but ignore the adaptive adjustment of transmitter’s illumination of the environment. As suggested by an idea called cognitive radar [2], which can adjust a transmitter’s illumination via dynamic closed-loop feedback, the investigation of adaptive waveforms design capable of enhancing target as well as suppressing clutter provides an efficient approach for improved radar performance. The superior performance of cognitive radar is achieved by merging knowledge-aided processing and full radar adaptivity into a real-time embedded computing architecture that use the information ahead [3]. The full radar adaptivity is also on both the transmit and receive sides. This method can significantly reduce clutter residue.

Waveform design is also important for radar detection where the radar might suffer performance degradation if waveform fails to meet radar requirement, like exhibiting time-varying features [4,5]. In existing work on waveform design, the clutter is almost assumed to follow a Gaussian distribution. One representative waveform design technology for target detection is signal-to-interference-plus-noise ratio (SINR) criterion as shown in [6,7], and an analytical solution for the optimal waveform spectrum based on SINR has been derived in [8]. However, the Gaussian assumption fails to handle clutter returns emerging non-Gaussian traits such as the increased occurrence of higher spikes or amplitudes. To overcome this condition, a compound-Gaussian (CG) process is introduced [9] and several models for the CG process have been developed [10,11]. Recently, in [12], the waveform optimization for improved detection under sea clutter has been considered and the transmitted waveform is adapted by minimizing its autocorrelation function where the clutter is estimated to be strong for clutter suppression where sea clutter returns are modelled as a CG process in this case, but this algorithm can only be used for detecting a point target. As for detecting the extended target, more adaptions are needed to be conducted like using several models for useful target echo [13], assuming the availability of secondary data free of useful signals [14], using a modified generalized likelihood ratio test without resorting to secondary data [15], and some combination [16].

In this paper, we attack a novel waveform design algorithm, namely, low range sidelobes of cognitive radar (LRSCR), for extended target detection in sea clutter backgrounds. Herein, the clutter is considered statistically as the CG model and the transmitted waveform is adapted based on the longer correlation characteristic of the texture. Moreover, an improved waveform design algorithm takes the output SINR improvement as well as peak sidelobes level (PSL) characteristics into consideration.

## 2. Problem Formulation

Consider waveform agility in a radar scene where there is an extended target embedded in CG sea clutter. The radar transmits *N* identical pulses in a dwell, and the incidence waveform agility is between dwells rather than on a pulse-to-pulse basis. A brief introduction of the cognitive radar system is necessary for the proposed waveform adaptation for extended target detection. The block diagram of cognitive radar via the feedback loop is illustrated in Figure 1, which can be described as follows: firstly, the transmitter transmits an optimal waveform, which is back scattered by the surrounding environment; secondly, the receiver estimates its environment parameters based on these returns; finally, the next transmitted signal is designed based on the current estimation results obtained by the feedback loop. For each closed loop operated by cognitive radar, the environment information can be estimated online and the so-estimated value in the current loop can be used as prior knowledge for waveform adjustment of the next loop. The prior knowledge includes the target impulse response s(t), sea clutter c(t) with covariance matrix Σ, and the noise v(t), which will be used in later calculations.

Subsequently, we focus on sea clutter statistics, which are key for adaptive waveform design. According to the CG model, *N*-dimensional clutter vector $c_{i}$ in the ith range cell can be given as:(1)$$c_{i}=\sqrt{\tau_{i}}\cdot \eta_{i}$$
where $\tau_{i}$ is the texture component, $\eta_{i}∼CN\left(0,\eta \right)$ is the speckle component with normalized covariance matrix η. Specifically, the texture represents the local clutter power with longer decorrelation time, which ensures that texture values for all clutter scatterers in a range cell remain identical during a dwell, as well as correlated between successive dwells. So in our optimization, the texture can be exploited to characterize the clutter ensemble over a dwell. Otherwise, the sea clutter impulse response series can be represented by the square root set of the texture at finite-number range cells.

Therefore, based on the CG characteristics of sea clutter, an investigation of intrapulse agility waveforms, which rely on the longer decorrelation time of the texture component, is feasible for improved extended target detection in sea clutter at least one dwell time by a cognitive radar system.

## 3. Waveform Design

### 3.1. The Expected Optimum Waveform Spectrum Design

Now consider the development of an adaptive waveform design algorithm for improved target detection under sea clutter, as shown in Figure 2. Let x(t) be a single pulse waveform having finite total energy $E_{x}$ with finite-duration *T* and finite-bandwidth *W*. The extended target echo is corrupted by clutter and receiver noise. Define X(f) as the Fourier transform of x(t) and $\sigma_{s}^{2}\left(f\right)$, $\sigma_{c}^{2}\left(f\right)$, $\sigma_{v}^{2}\left(f\right)$ as the spectral variance of the target, clutter and noise impulse response respectively, which has been obtained in the cognitive radar loop as prior knowledge. Note that the target spectral mean is not equal to zero. The receiving filter frequency response is H(f). The output SINR at instant $t_{0}$ is:(2)$${\left(SINR\right)}_{t_{0}}=\frac{\int_{W}{\left|X\left(f\right)\right|}^{2}\sigma_{s}^{2}\left(f\right){\left|H\left(f\right)\right|}^{2}df}{\int_{W}L\left(f\right){\left|H\left(f\right)\right|}^{2}df}$$
where $L\left(f\right)={\left|X\left(f\right)\right|}^{2}\sigma_{c}^{2}\left(f\right)+\sigma_{v}^{2}\left(f\right)$ According to Cauchy-Schwartz inequality, when a matched filter satisfies:(3)$$\left|H\left(f\right)\right|=k\frac{\sqrt{\sigma_{s}^{2}\left(f\right)}\left|X\left(f\right)\right|}{{\left|X\left(f\right)\right|}^{2}\sigma_{c}^{2}\left(f\right)+\sigma_{v}^{2}\left(f\right)}$$
where *k* is a nonzero constant, the output SINR achieves the maximum at instant $t_{0}$ as:(4)$${\left(SINR\right)}_{t_{0}\;max}=\int_{W}\frac{{\left|X\left(f\right)\right|}^{2}\sigma_{s}^{2}\left(f\right)}{{\left|X\left(f\right)\right|}^{2}\sigma_{c}^{2}\left(f\right)+\sigma_{v}^{2}\left(f\right)}df$$

Discretizing the frequency band into *K* sub-bands, the waveform optimization problem of maximization of the output SINR over the waveform spectrum can be written in the following terms:(5)$$\begin{array}{cc} \max_{|X\left(f_{k}\right){|}^{2}}= & \sum_{k=0}^{K-1}\frac{|X\left(f_{k}\right){|}^{2}\sigma_{s}^{2}\left(f_{k}\right)}{|X\left(f_{k}\right){|}^{2}\sigma_{c}^{2}\left(f_{k}\right)+\sigma_{v}^{2}\left(f_{k}\right)}\\ s.t. & \sum_{k=0}^{K-1}{\left|X\left(f_{k}\right)\right|}^{2}=E_{x}\\ & |X\left(f_{k}\right){|}^{2}\geq 0,\;k=0,1,\ldots ,K-1 \end{array}$$

So the expected optimum waveform spectrum can be obtained by solving Equation (Equation 5) as:(6)$$|X\left(f_{k}\right){|}^{2}=max\left[0,\frac{\sqrt{\sigma_{s}^{2}\left(f_{k}\right)\sigma_{v}^{2}\left(f_{k}\right)}}{\sqrt{\sigma_{c}^{2}\left(f_{k}\right)}}\left(A-\sqrt{\frac{\sigma_{v}^{2}\left(f_{k}\right)}{\sigma_{s}^{2}\left(f_{k}\right)}}\right)\right],\;k=0,1,\ldots ,K-1$$
where *A* is a constant determined by the transmitted waveform energy $E_{x}$ shown in Equation (Equation 7) [8].
(7)$$\sum_{k=0}^{K-1}max\left[0,\frac{\sqrt{\sigma_{s}^{2}\left(f_{k}\right)\sigma_{v}^{2}\left(f_{k}\right)}}{\sqrt{\sigma_{c}^{2}\left(f_{k}\right)}}\left(A-\sqrt{\frac{\sigma_{v}^{2}\left(f_{k}\right)}{\sigma_{s}^{2}\left(f_{k}\right)}}\right)\right]=E_{x}$$

Denoted by the power spectral density (PSD) $P_{opt}=\left[\right|X_{opt}\left(f_{0}\right){|}^{2},|X_{opt}\left(f_{1}\right){|}^{2},\ldots ,|X_{opt}\left(f_{K-1}\right){{|}^{2}]}^{T}$, where ${\left(\cdot \right)}^{T}$ denotes transpose, the so-obtained waveform is the optimum solution providing the best detection performance, without any control of the waveform shapes or forms.

### 3.2. Design of the LRSCR Waveform

To make full use of the transmitter power, a phase-modulated (PM) waveform $x={\left[x_{0},x_{1},\ldots ,x_{N-1}\right]}^{T}$ is adopted in discrete time sampling given by $x=ae^{j\phi }$, where *a* is a constant determined by the transmitter power, and ϕ denotes the phase vector. Correspondingly, the spectrum of the designed waveform can be expressed as:(8)$$X\left(k\right)=\sum_{n=0}^{N-1}x_{n}e^{-j\frac{2\pi nk}{K}},\;k=0,1,\ldots ,K-1$$

To clarify Equations (Equation 5) and (Equation 8), both X(K) and $X(f_{K})$ represent the spectrum but the former one is for the discrete time sampling and later one is for the continues signal. The proposed LRSCR waveform for improved target detection performance can be designed subject to the key items as: (1) maximization of the output SINR, and (2) lower range sidelobes level.

Denoted by PSD $P_{opt}={\left[|X\left(0\right)\right|}^{2}{,|X\left(1\right)|}^{2}{,\ldots ,|X\left(K-1\right)|}^{2}{]}^{T}$, the designed waveform subjected to item (1), whose goal is to find transmitted waveform that maximizes the output SINR, can be optimized equivalently by minimizing the value of mean-square error as:(9)$$\min_{x}\left|\right|P_{x}-P_{opt}{\left|\right|}_{2}^{2}$$
where $\left|\right|\cdot {\left|\right|}_{2}^{2}$ denotes Euclidean 2-norm.

Note now that the optimization problem (9) only considers maximizing the output SINR. One serious problem is that these SINR-based waveforms are always deficient in range sidelobes performance. In most real maritime environments, high sidelobes will result in increasing false alarm rates, or masking a weak reflection from another target. For this reason, there is a key requirement to impose another sidelobes constraint on the optimal waveform. The waveform autocorrelation function can be expressed as:(10)$$R_{x}\left(l\right)=\left\{\begin{array}{c} \sum_{n=l}^{M-1}x_{m}x_{m-l}^{*},\;l=0,1,\ldots ,M-1\\ R_{x}^{*}\left(-l\right),\;l=-\left(M-1\right),-\left(M-2\right),\ldots ,-1 \end{array}\right.$$
where ${\left(\cdot \right)}^{*}$ denotes conjugate. Consider PSL for sidelobes performance evaluation, the cost function of minimizing the PSL can be given as:(11)$$\min_{x}\max_{l=-(M-1)\;l\neq 0}^{M-1}\left\{\right|R_{x}\left(l\right)\left|\right\}$$

Combining Equations (Equation 9) and (Equation 11) provides waveform optimization problem as:(12)$$\min_{x}\left\{\beta |\right|P_{x}-P_{opt}{\left|\right|}_{2}^{2}+\left(1-\beta \right)\max_{l=-(M-1)\;l\neq 0}^{M-1}|R_{x}\left(l\right)\left|\right\}$$
where β∈(0,1) denotes weight parameter. Observe that the problem (Equation 11) is a multidimensional nonlinear optimization and can be solved by genetic algorithm. The genetic algorithm used in this paper followed the same process in a previous study [17]. The major complexity of the proposed algorithm is restricted by the genetic algorithm that searches for the best solution. The more iterations it runs, the better results it can get. Therefore, it is a trade-off between computation time and performance. After this optimum waveform is transmitted, its clutter suppression performance can be measured by the output SINR given as:(13)$$SINR_{out}=\frac{max(|y_{s}{\left(t\right)}^{2}|)}{var\left(y_{c}\left(t\right)\right)+var\left(y_{v}\left(t\right)\right)}$$
where $y_{s}\left(t\right)$, $y_{c}\left(t\right)$, and $y_{v}\left(t\right)$ denote the filter outputs of the target, clutter and noise respectively, var(·) denotes variance.

## 4. Performance Assessment

We present a Monte Carlo simulation to demonstrate the detection performance of the proposed LRSCR waveform, also in comparison to the SINR-based waveform in Equation (Equation 9) without sidelobes constraints (denoted by SINR-based waveform) and to the linear frequency modulated (LFM) waveform. Firstly, the expected optimum waveform spectrum is obtained, then the detection characteristics of these three signals are compared; both the output SINR improvement and PSL characteristics are under consideration for waveform evaluation.

### 4.1. Simulation Setup

The experimental radar system parameters are as follows: radar carrier frequency 1 GHz, pulse number *N* in a dwell is 10, pulse duration 1 μs, bandwidth 100 MHz, sampling rate 200 MHz; each transmitted pulse has unit energy. Consider a nonfluctuating extended target with impulse response $h_{s}\left(t\right)=\sum_{l=0}^{L-1}\alpha_{l}\delta \left(t-l\right)$ is distributed in L=3 range cells with l=20,40,60, where $\alpha_{l}$ denotes target amplitude. Suppose sea clutter is modelled as CG process with covariance matrix Σ given by ${\left[\Sigma \right]}_{ij}=0.9^{\left|i-j\right|}$, where 1≤i,j≤N are the pulses index in a dwell, and texture $\tau_{i}s$ followed a Gamma distribution with shape parameter *v* and power mean value $\sigma_{1}^{2}$, i.e., $f\left(\tau_{k}\right)={\left(\frac{v}{\sigma_{1}^{2}}\right)}^{v}\tau_{k}^{v-1}e^{\frac{-v\tau_{k}}{\sigma_{1}^{2}}}/\Gamma \left(v\right)$, where Γ(·) denotes the Gamma function. The noise is modelled as white noise generated as $v∼CN_{N}\left(0,\sigma_{2}^{2}I_{N}\right)$. Define the input signal-to-clutter ratio (SCR) and clutter-to-noise ratio (CNR) as $SCR_{in}=max(|h_{s}\left(n\right){|}^{2})/\sigma_{1}^{2},\;CNR_{in}=\sigma_{1}^{2}/\sigma_{2}^{2}$. Here the input SCR and CNR are set to 20 dB and 10 dB respectively. Finally, denoted by , the vector of matched-filtered outputs in the th range cell is detected by an adaptive detector given by $\lambda =|p^{H}\Sigma^{-1}y_{i}{|}^{2}/\left(p^{H}\Sigma^{-1}p\right)\left(y_{i}^{H}\Sigma^{-1}y_{i}\right)$, where *p* denotes target steering vector $p={\left[1,e^{j\theta },\ldots ,e^{j(N-1)\theta }\right]}^{T}/\sqrt{N}$ and θ denotes a constant phase shifting, $y_{i}$ is the receiver output of the ith range cell under test, and ${\left(\cdot \right)}^{H}$ denotes conjugate transpose. In a dwell period, a plot of typical responses of the extended target and sea clutter is shown in Figure 3.

### 4.2. The Expected Optimum Waveform Spectrum

Based on the maximum output SINR rule, the expected optimum waveform spectrum can be obtained by Equation (Equation 6). Figure 4 demonstrates the spectral variance of the expected optimum waveform, target and sea clutter. As can be seen from Figure 4, the expected transmitted waveform concentrated most of its energy into frequency components where the target is strong and the clutter is weak, while dispersing less energy to both weak target and strong clutter frequency bands. Obviously, the expected optimum waveform spreads the majority of its energy into the regions where the target’s frequency response is larger relative to the clutter’s in order to enhance the output SINR for improved target detection performance.

### 4.3. Detection Performance Analysis

Based on the expected optimum spectrum solution in Figure 4, the SINR-based waveform and proposed LRSCR waveform (β=0.5) can be obtained by solving Equations (Equation 9) and (Equation 12), respectively. Evidently, as demonstrated in Figure 5, the spectrum of SINR-based signal (Figure 5a left) agrees well with the expected optimum spectrum, but showing high autocorrelation function sidelobes (Figure 5a right); and LRSCR waveform shows low autocorrelation function sidelobes (Figure 5b right), but at the expense of certain waveform spectrum matching loss with the expected optimum spectrum (Figure 5b left). Furthermore, the three signals, i.e., the LRSCR, SINR-based and LFM waveforms, are illuminated into the scenario depicted in Figure 3. A plot of range profiles for these three waveforms is illustrated in Table 1, and Figure 6 on detection performance lists the corresponding output SINR and PSL.

As expected, in Figure 6a the LRSCR waveform with output SINR 21.1 dB and PSL −23.8 dB can enhance the energy in interesting target cells, as well as restrain the energy in undesired clutter cells for improved target detention. In Figure 6b, the SINR-based waveform also provides a high output SINR 21.0 dB, but the presence of high range sidelobes −1.2 dB of this waveform results in some false targets, which necessarily increases false alarm rate and significantly impacts radar detection performance. In Figure 6c, the common LFM waveform with output SINR 18.9 dB and PSL −13.5 dB exhibits a moderate behavior between the LRSCR and SINR-based waveforms.

Figure 7 illustrates the probability of detection (Pd) versus the probability of false alarm (Pfa) for the LRSCR, SINR-based and LFM waveforms. It is evident that the LRSCR signal provides a detection performance better than that of the other two signals as confirmed by Figure 6, while the SINR-based waveform presents increasing false-alarm rate. Apparently, the LRSCR signal can obtain high output SINR for clutter restraint and low range sidelobes for false-alarm rate reduction, both of which will further improve target detection performance in clutter.

## 5. Conclusions

Adaptive waveform design for extended target detection in cognitive radar frameworks is addressed under CG sea clutter, proposing a novel design algorithm for maximization of output SINR subject to low sidelobes levels. The longer decorrelation time of the texture component within successive dwell time ensures the feasibility for intrapulse agility waveforms in cognitive radars to improve target detection performance in a maritime environment. Based on the expected waveform spectrum over the maximization of the output SINR, an optimization method which designs transmitted waveforms minimizing the mean square error in locating the expected waveform spectrum in conjunction with restraining range sidelobes is performed to yield optimal waveforms that have both good clutter suppression performance and low sidelobes levels. Simulation results illustrated that the proposed waveforms can guarantee good clutter suppression performance as well as sidelobes characteristics, and achieve a better detection performance than existing waveforms.

## Figures and Tables

**Figure 1 sensors-19-03957-f001:**
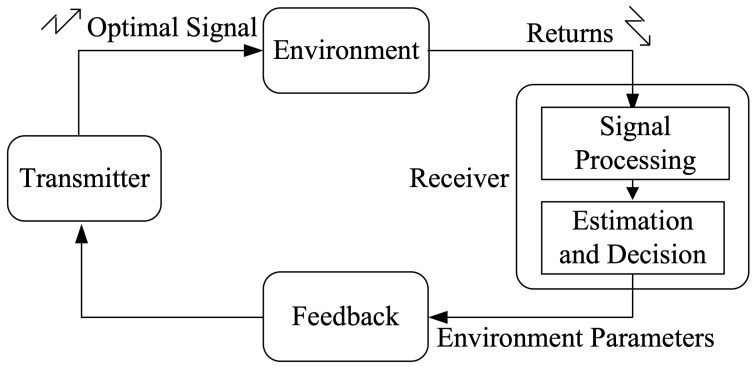
Black diagram of cognitive radar via the feedback loop.

**Figure 2 sensors-19-03957-f002:**
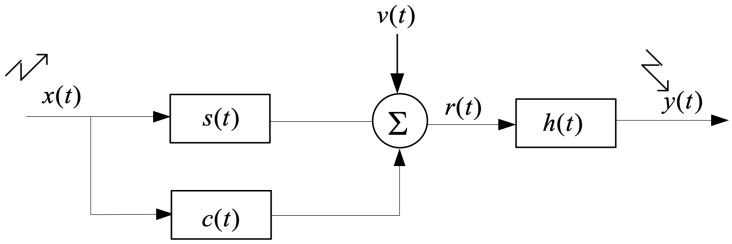
Model of the received signal. x(t) is the transmitted signal, s(t) is the extend target impulse response with the spectral variance $\sigma_{s}^{2}\left(f\right)$, c(t) represents the clutter with the spectral variance $\sigma_{c}^{2}\left(f\right)$, v(t) represents ambient noise with the spectral variance $\sigma_{v}^{2}\left(f\right)$, r(t) is the received baseband signal, h(t) is the receiver-filter, and y(t) is the receiver output.

**Figure 3 sensors-19-03957-f003:**
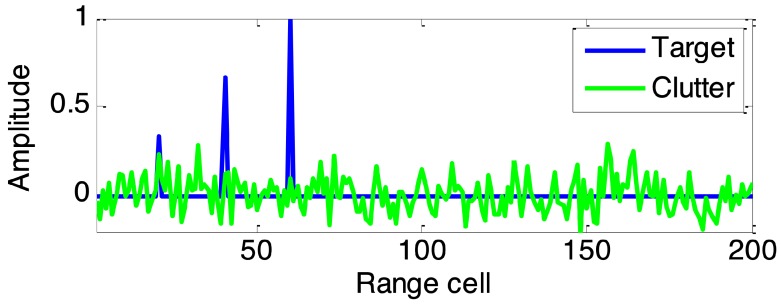
The impulse responses of the extended target and sea clutter.

**Figure 4 sensors-19-03957-f004:**
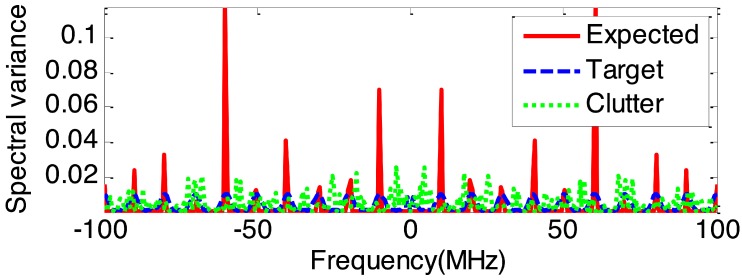
The spectral variance of the expected optimum waveform, target and sea clutter.

**Figure 5 sensors-19-03957-f005:**
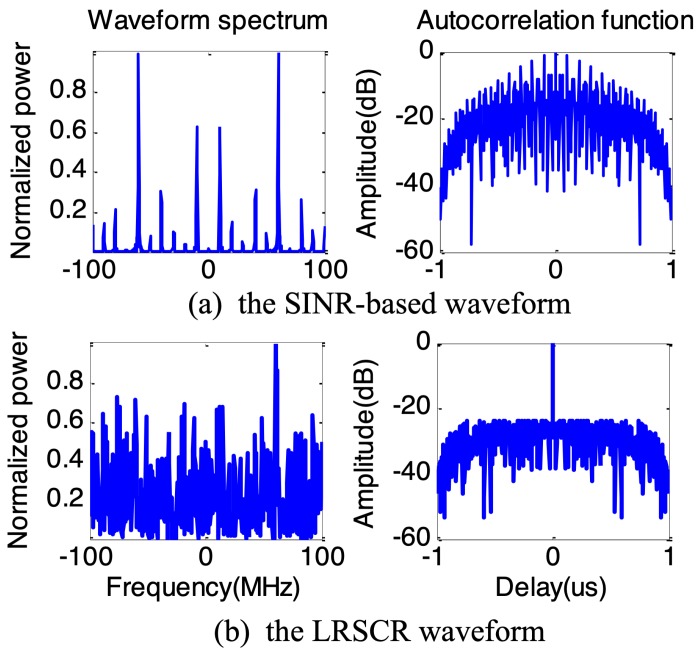
The waveform spectrum and autocorrelation function.

**Figure 6 sensors-19-03957-f006:**
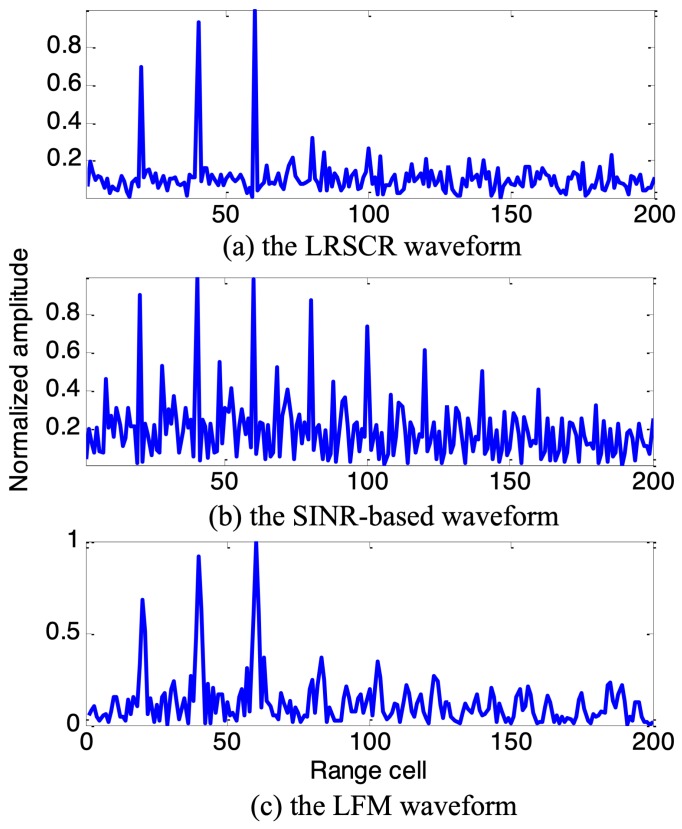
The range profile for considered waveforms.

**Figure 7 sensors-19-03957-f007:**
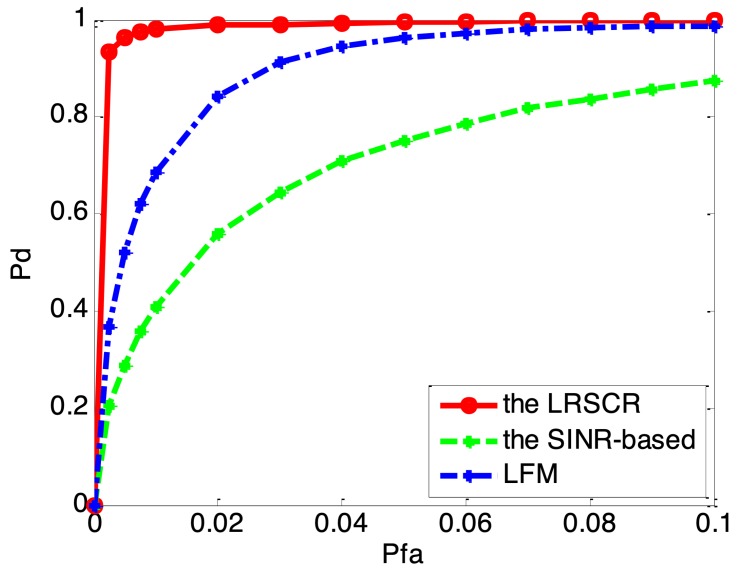
Pd versus Pfa for considered waveforms.

**Table 1 sensors-19-03957-t001:** Performance comparison for considered waveforms.

Waveforms	*SINR_out_*	PSL (dB)
LRSCR	21.1	−23.8
SINR-based	21.0	−1.2
LFM	18.9	−13.5

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
