# Peer review of "Waveform Design for Improved Detection of Extended Targets in Sea Clutter"

_sensors, 2019, doi:10.3390/s19183957_

Round 1

Reviewer 1 Report

The paper proposes waveform design for detection of an extended target in compound Gaussian sea clutter. The paper first uses the SINR as an objective function to solve for an SINR-based optimal waveform. Then design the waveform for a dwell by approximating the desired waveform to this SINR-based optimal waveform, with the consideration of the sidelobes.  The optimal solution is obtained by a genetic algorithm. The performance is compared with those use LFM waveform and SINR-based optimal waveform. Here are some comments to improve the paper:

1) In figure 1 and the discussion nearby, estimating environment parameters are mentioned. However, the authors do not provide a clear connection to this part in the later sections. I guess that the authors refer to the parameters that are implicitly assumed to be known in the waveform design, for example, $\sigma_s(f)$,  $\sigma_c(f)$, and $\sigma_v(f)$ in equation (2). Please clarify.

2) In practice, how do you estimate $\sigma_s(f)$,  $\sigma_c(f)$, and $\sigma_v(f)$, which are assumed to be known in the paper? Also, please specify in the paper the parameters that are implicitly assumed to be known.

3) In section 3, to make the presentation clearer, I recommend that the authors provide a received signal model that includes target impulse, clutter, and noise. This model should have the waveform x[n] and the extend target impulse response h[n] in section 4.1. Connect this model to the parameters the authors define in the paper. Specify the relation between $\sigma_s(f)$ and $h[n]$, and mention what $p$ and $y$ are nearby.

4) Add references in section 3 for the analysis shown there and the water-filling solution in (6) and (7).

5) Less than or equal sign should be $=$ in equation (7). This energy should be attained.

6) Add a reference to the genetic algorithm that the authors use in the paper.

7) Clarify the relation between $X(K)$ in (8) and $X(f_k)$ in (4).

8) Clarify $i$ and $j$  in section 4.1 for $[\Sigma]_{ij}$, which is covariance matrix.

9) The reviewer would recommend that the authors fix some grammatical problems and improve the presentation, for example:

    Line 34, “the implied waveform agility” is confusing;

    Line 44, this line should not be indented;

    In section 3.1, “undersea clutter” -> “under sea clutter”.

Author Response

Dear Reviewer 1#

RE: Ms. Waveform Design for Improved Detection of Extended Target in Sea Clutter

We thank you for your constructive comments and opportunity to improve our manuscript for publication inSensors.

We here address each query in turn in the attached pdf file

Reviewer 2 Report

The present paper deals with the problem of radar waveform design in compound Gaussian sea clutter for extended targets. The idea consists in design the waveform optimizing the SINR imposing also a constraint on the sidelobe of the waveform autocorrelation function.

in the Reviewer opinion, the idea is interesting, and the results are encouraging. Moreover, the paper is well written and easy to follow. However, the Reviewer suggests the following modification/corrections to improve the quality of manuscript before final acceptation.

First of all, as to eqs. (6) and (7), please enlarge the brackets. Moreover, in eqs. (11) and (12) Rxl should be Rx(l).

A discussion on the computational complexity of the algorithm is needed.

Finally, the Introduction is meager and must be expanded with the discussion and citation of other related papers (actually the references are only 6) such as those listed in the following:

Radar waveform design [1]-[4];

Compound sea-clutter applications (detection, covariance estimation, …) [5]-[7];

Extended targets detection [8]-[10];

Cognitive radar [11]-[13].

[1] K. Gerlach, “Thinned spectrum ultrawideband waveforms using stepped-frequency polyphase codes”, IEEE Transactions on Aerospace and Electronic Systems, vol. 34, no. 4, pp. 1356-1361, Oct. 1998.

[2] S. D. Blunt, J. G. Metcalf, C. R. Biggs, and E. Perrins, “Performance characteristics and metrics for intra-pulse radar-embedded communication”, IEEE Journal on Selected Areas in Communications, vol. 29, no. 10, pp. 2057-2066, Dec. 2011.

[3] M. Wicks, E. Mokole, S. Blunt, R. Schneible, and V. Amuso, “Principles of Waveform Diversity and Design”, Raleigh NC: SciTech Publishing 2011.

[4] A. Aubry, V. Carotenuto, A. D. Maio, A. Farina and L. Pallotta, “Optimization theory-based radar waveform design for spectrally dense environments”, IEEE Aerospace and Electronic Systems Magazine, vol. 31, no. 12, pp. 14-25, December 2016.

[5] E. Conte, M. Lops, and G. Ricci, “Asymptotically optimum radar detection in compound-Gaussian clutter”, IEEE Trans. on Aerospace and Electronic Systems, vol. 31, no. 2, pp. 617-625, 1995.

[6] F. Pascal, Y. Chitour, J.-P. Ovarlez, et al. "Covariance structure maximum-likelihood estimates in compound Gaussian noise: existence and algorithm analysis", IEEE Trans. on Signal Processing, vol. 56, no. 1, pp. 34-48, 2008.

[7] G. Cui, N. Li, L. Pallotta, G. Foglia and L. Kong, "Geometric barycenters for covariance estimation in compound-Gaussian clutter," IET Radar, Sonar & Navigation, vol. 11, no. 3, pp. 404-409, 3 2017.

[8] K. Gerlach and M. J. Steiner, "Adaptive detection of range distributed targets", IEEE Trans. on Signal Processing, vol. 47, no. 7, pp. 1844-1851, July 1999.

[9] F. Bandiera, D. Orlando, and G. Ricci, "CFAR detection of extended and multiple point-like targets without assignment of secondary data", IEEE Signal Processing Letters, vol. 13, no. 4, pp. 240-243, April 2006.

[10] A. Aubry, A. De Maio, L. Pallotta, and A. Farina, "Radar Detection of Distributed Targets in Homogeneous Interference Whose Inverse Covariance Structure is Defined via Unitary Invariant Functions," IEEE Trans. on Signal Processing, vol. 61, no. 20, pp. 4949-4961, Oct.15, 2013.

[11] J. R. Guerci, “Cognitive Radar: The Knowledge-Aided Fully Adaptive Approach”, Norwood MA: Artech House 2010.

[12] A. Aubry, V. Carotenuto, A. De Maio and L. Pallotta, “High Range Resolution Profile Estimation via a Cognitive Stepped Frequency Technique”, IEEE Transactions on Aerospace and Electronic Systems, vol. 55, no. 1, pp. 444-458, February 2019.

[13] P. Addabbo, A. Aubry, A. De Maio, L. Pallotta and S. L. Ullo, "HRR profile estimation using SLIM," in IET Radar, Sonar & Navigation, vol. 13, no. 4, pp. 512-521, 4 2019.

Author Response

Dear Reviewer 2#

RE: Ms. Waveform Design for Improved Detection of Extended Target in Sea Clutter

We thank you for your constructive comments and opportunity to improve our manuscript for publication inSensors.

We here address each query in turn in the attached pdf file

Round 2

Reviewer 1 Report

The authors have well addressed my comments and made changes in the paper.